



**Examination of Brown Carbon Absorption from Wildfires in the Western U.S. During the**
**WE-CAN Study**

Amy P. Sullivan[1], Rudra P. Pokhrel[2,*], Yingjie Shen[2], Shane M. Murphy[2], Darin W. Toohey[3],
Teresa Campos[4], Jakob Lindaas[1], Emily V. Fischer[1], and Jeffrey L. Collett, Jr.[1]
[1]Colorado State University, Department of Atmospheric Science, Fort Collins, Colorado 80523
[2]University of Wyoming, Department of Atmospheric Science, Laramie, WY 82071
[3]University of Colorado – Boulder, Department of Atmospheric and Oceanic Sciences, Boulder,
CO 80309
[4]National Center for Atmospheric Research, Atmospheric Chemistry Division, Boulder, CO
*now at Cooperative Institute for Research in Environmental Sciences, University of Colorado,
Boulder, CO 80309 and NOAA Chemical Science Laboratory, Boulder, CO 80305
Corresponding author: Amy P. Sullivan, sullivan@atmos.colostate.edu





**Abstract**
Light absorbing organic carbon, or brown carbon (BrC), can be a significant contributor
to the visible light absorption budget. However, the sources of BrC and the contributions of BrC
to light absorption are not well understood. Biomass burning is thought to be a major source of
BrC. Therefore, as part of the WE-CAN (Western Wildfire Experiment for Cloud Chemistry,
Aerosol Absorption and Nitrogen) Study BrC absorption data was collected aboard the
NSF/NCAR C-130 aircraft as it intercepted smoke from wildfires in the Western U.S. in July-
August 2018. BrC absorption measurements were obtained in near real-time using two
techniques. The first coupled a Particle-into-Liquid Sampler (PILS) with a Liquid Waveguide
Capillary Cell and a Total Organic Carbon analyzer for measurements of water-soluble BrC
absorption and WSOC (water-soluble organic carbon). The second employed a custom-built
Photoacoustic Aerosol Absorption Spectrometer (PAS) to measure total absorption at 405 and
660 nm. The PAS BrC absorption at 405 nm (PAS total Abs 405 BrC) was calculated by
assuming the absorption determined by the PAS at 660 nm was equivalent to the black carbon
(BC) absorption and the BC aerosol absorption Ångström exponent was 1. Data from the PILS
and PAS were combined to investigate the water-soluble vs. total BrC absorption at 405 nm in
the various wildfire plumes sampled during WE-CAN. WSOC, PILS water-soluble Abs 405,
and PAS total Abs 405 tracked each other in and out of the smoke plumes. BrC absorption was
correlated with WSOC ($R^2$ value for PAS = 0.42 and PILS = 0.60) and CO (carbon monoxide)
($R^2$ value for PAS = 0.76 and PILS = 0.55) for all wildfires sampled. The PILS water-soluble
Abs 405 was corrected for the non-water-soluble fraction of the aerosol using the calculated
UHSAS (Ultra-High-Sensitivity Aerosol Spectrometer) aerosol mass. The corrected PILS water-
soluble Abs 405 showed good closure with the PAS total Abs 405 BrC with a factor of ~1.5 to 2
difference. This difference was explained by particle vs. bulk solution absorption measured by
the PAS vs. PILS, respectively, and confirmed by Mie Theory calculations. During WE-CAN,
~45% (ranging from 31% to 65%) of the BrC absorption was observed to be due to water-soluble
species. The ratio of BrC absorption to WSOC or ΔCO showed no clear dependence on fire
dynamics or the time since emission over 9 h.





## 1. Introduction

Organic compounds can comprise a large fraction of PM (particulate matter) mass [*Kanakidou et al.*, 2005; *Zhang et al.*, 2007]. Organic carbon can be directly emitted or formed in the atmosphere from a variety of sources. This leads to organic aerosol particles composed of a number of compounds that range from insoluble to highly water-soluble and that can scatter or absorb light [*Jacobson et al.*, 2000; *Saxena and Hildemann*, 1996, and references therein].

The portion of organic carbon that is light-absorbing has been referred to as brown carbon (BrC) due to its yellow or brown color when concentrated, and it is likely to be a significant contributor to the visible light-absorption budget [*Andreae and Gelencsér*, 2006]. Recent modeling studies have predicted a non-negligible effect on the Earth's radiation balance from BrC [*Feng et al.*, 2013; *Zhang et al.*, 2017; *Zhang et al.*, 2020]. Global measurements have shown that BrC can contribute up to 48% of the overall warming effect by absorbing carbonaceous aerosols [*Zeng et al.*, 2020]. BrC may also suppress photolysis rates of some chemical reactions, including decreasing surface ozone concentrations in certain locations, due to its ability to absorb at ultraviolet wavelengths [*Jo et al.*, 2016]. Some portion of BrC is likely composed of toxins, such as nitro- and oxy-aromatic species, suggesting BrC could also have health impacts [*Desyaterik et al.*, 2013; *Verma et al.*, 2015; *Zhang et al.*, 2013]. BrC itself is thought to have both primary and secondary sources. Particles from biomass burning or incomplete combustion of fossil fuels generally contain significant amounts of BrC (e.g., [*Chakrabarty et al.*, 2010; *Hoffer et al.*, 2006; *Kirchstetter et al.*, 2004; *Kirchstetter and Thatcher*, 2012; *Lack et al.*, 2012; *Lukács et al.*, 2007]). Laboratory studies have observed production of BrC from a number of formation processes. This has included heterogenous reactions of isoprene on acidic aerosol particles, a variety of aqueous-phase reactions, and reactions of organic compounds in acidic solutions (e.g., [*Hoffer et al.*, 2006; *Limbeck et al.*, 2013; *Sareen et al.*, 2010; *Updyke et al.*, 2012]). However, there is still limited information on the contribution of BrC to total light absorption and the sources of BrC as there are few ambient measurements.

Total absorption measurements (black carbon (BC) + BrC) at multiple wavelengths can be used to determined BrC absorption due to the strong wavelength dependence of BrC. This requires the assumptions that: (1) the absorption Ångström exponent (AAE) for BC is known, (2) AAE is constant with wavelength, and (3) BrC does not absorb at longer wavelengths. The AAE for BC is well constrained at 1 in the visible and near-infrared wavelengths [*Moosmüller et al.*, 2009]. The BrC absorption at other wavelengths is then found by difference from the extrapolated BC AAE [*Lack and Langridge*, 2013; *Mohr et al.*, 2013]. This approach can be applied to any technique that measures absorption at multiple wavelengths, including photoacoustic spectroscopy (PAS).

BrC can also be quantified by isolating the BrC chromophores by extraction of particles in solvents, such as water or methanol, in order to separate them from the insoluble BC and then measuring the light absorption of the soluble organic chromophores [*Hecobian et al.*, 2010]. This is the only direct method to separate and quantify BrC as the light absorption from liquid extracts does not suffer from interferences by BC as they can be isolated by dissolution. A spectrophotometer with an UV/Vis (ultraviolet/visible) light source can provide high spectral resolution over a wide wavelength range from 200 to 800 nm. In addition, when coupled with a long-path liquid waveguide capillary absorption cell (LWCC), it also provides a highly sensitive measurement. This technique can be used off-line with filters or on-line with an aerosol





collection device such as Particle-into-Liquid Sampler (PILS) (e.g., [*Hecobian et al.*, 2010; *Liu*
*et al.*, 2013, 2014, 2015; *Zhang et al.*, 2011, 2013].
Here we report BrC absorption data from a PAS and PILS-LWCC-TOC system to
compare total vs. water-soluble BrC absorption in wildfire smoke.  Data are from smoke plume
penetrations during the Western Wildfire Experiment for Cloud Chemistry, Aerosol Absorption
and Nitrogen (WE-CAN), an aircraft-based study focused on investigating the chemistry and
transformation of emissions from wildfires in the western U.S.  We examine the relationship
between the BrC absorption and species known to be from biomass burning.  We discuss how
parameters such as aging and fire dynamics might influence BrC absorption from wildfires.
**2.  Methods**
**2.1.  The Airborne Mission**
The WE-CAN Campaign was a multi-investigator study conducted aboard the National
Science Foundation/National Center for Atmospheric Research (NSF/NCAR) C-130 aircraft.
The C-130 was operated out of Boise, ID from Jul. 20 to Aug. 31, 2018.  A suite of instruments
was deployed for measurements of aerosol and trace gas composition.  A total of 16 research
flights sampled wildfire smoke over the western U.S. to characterize emissions, mixing,
chemical transformations, and transport.  Figure 1 presents a map of the flight transects and
locations of the wildfires sampled.  (We exclude Flight RF14, which was conducted off the coast
of CA to sample a stratus deck impacted by smoke, and Flight RF16, which consisted of an
intercomparison performed near Boise between WE-CAN and BB-FLUX (Biomass Burning
Flux Measurements of Trace Gases and Aerosols) common measurements.)  More information
on each wildfire including the type of fuel consumed is available in the WE-CAN Field Catalog
(catalog.eol.ucar.edu/we-can).  WE-CAN sampled both fresh and aged (for Flights RF05 and
RF08 along with parts of Flights RF07 and RF13) emissions from smoke for wildfires burning in
CA, OR, WA, ID, MT, UT, and NV.  The general sampling strategy was to circle the wildfire at
the source and then follow the smoke downwind using a multiple transect search and rescue
pattern to examine smoke evolution.  Typically, wildfire smoke plumes were sampled in the free
troposphere between 3-5 km during early afternoon to evening periods (20:00 to 02:00 UTC or
14:00 to 20:00 LT).  Flight RF08 and part of Flight RF07 were exceptions as the aircraft sampled
the boundary layer (below 2 km) over the Central Valley of CA.
**2.2.  Particle Collection**
During WE-CAN, we deployed two Particle-into-Liquid Sampler (PILS) systems.  A
PILS is an aerosol collection device that continuously collects ambient particles into purified
water [*Orsini et al.*, 2003].  After particles are grown inside the body of the PILS by water
condensation in a supersaturated water vapor environment, formed through mixing the ambient
air sample with saturated air (100% relative humidity) at higher temperature, the particles are
collected by an impactor.  The impactor plate is continually washed off by a flow of liquid
passed over the impactor, providing a liquid sample containing dissolved aerosol particles which
can be analyzed by various methods.  Each PILS system sampled from a Submicron Aerosol
Inlet (SMAI) [*Craig et al.*, 2013a, 2013b, 2014; *Moharreri et al.*, 2014] mounted to the belly of
the NSF/NCAR C-130.  The size-cut for each PILS was provided by a nonrotating MOUDI
impactor stage with a 50% transmission efficiency of 1 μm aerodynamic diameter (i.e., $PM_1$) at 1
atmosphere ambient pressure [*Marple et al.*, 1991].  The flowrate of 15 LPM was sampled by

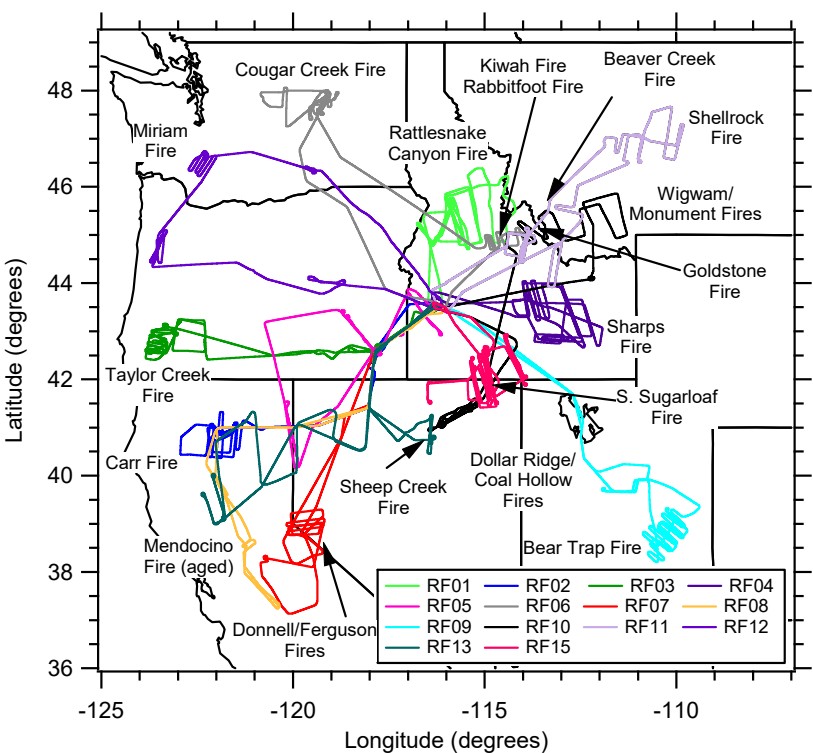

Figure 1. Map showing the flight paths and locations of the wildfires sampled during WE-CAN used in this analysis.

each PILS through the inlet and MOUDI stage. An activated carbon parallel plate denuder
[*Eatough et al.*, 1993] was situated upstream of both PILS to remove organic gases. In addition,
for PILS2 two honeycomb denuders coated with sodium carbonate and phosphorous acid were
used to remove inorganic acidic and basic gases in order to limit possible positive artifacts from
dissolving in the PILS collection liquid. PILS 1 was connected to a LWCC (liquid waveguide
capillary cell) and TOC (Total Organic Carbon) Analyzer for near real-time measurement of
water-soluble BrC (Brown Carbon) absorption and WSOC (water-soluble organic carbon),
respectively. PILS2 was coupled to a Bretchel fraction collector system [*Sorooshian et al.*,
2006] to provide liquid samples for additional off-line analysis.
For PILS 1, a valve upstream of the PILS was manually closed periodically for 10 min
forcing the airflow through a Teflon filter allowing for a measurement of the background in near
real-time. The liquid sample obtained from PILS1 was then pushed through a 0.2 μm PTFE
liquid filter at a flowrate of 1.2 mL/min by a set of syringe pumps with 1 mL syringes to ensure
any insoluble particles were removed before passing through the LWCC and TOC Analyzer.
A LWCC with a 2.5 m path-length (World Precision Instruments, Sarasota, FL) was
employed. An absorption spectrometer (FLAME-T-UV-VIS, Ocean Optics, Largo, FL) and dual
deuterium and tungsten halogen light source (DH-mini, Ocean Optics, Largo, FL) were coupled



to the LWCC via fiber optic cables.  The Oceanview Spectroscopy Software was used to record
absorption spectra over a range from 200 to 800 nm.  In this paper we present the absorption
determined at 365 and 405 nm.  This wavelength dependent absorption was calculated following
the method outlined in *Hecobian et al.* [2010].  A 16 s integrated measurement of water-soluble
absorption with a limit of detection (LOD) of 0.1 Mm$^{-1}$ was obtained.
A Sievers Model M9 Portable TOC Analyzer (Suez Waters Analytical Instruments,
Boulder, CO) was used for the WSOC measurement.  This analyzer converts organic carbon in
the liquid sample to carbon dioxide using chemical oxidation with ammonium persulfate and
ultraviolet light.  The carbon dioxide formed is then measured by conductivity.  The amount of
OC present in the sample is proportional to the increase in conductivity observed.  The analyzer
was run in turbo mode providing a 4 s integrated measurement of WSOC with a LOD of 0.1 μg
C/m$^3$.
For PILS2, a valve upstream of the PILS was manually closed periodically for 10 min
forcing the airflow through a hepa filter allowing for measurement of the background in near
real-time.  The liquid sample obtained from PILS2 was pushed into the fraction collector vials at
a flowrate of 0.65 mL/min by a peristaltic pump for collection of ~1.2 mL of liquid sample per
vial.  Each fraction collector carousel holds 72 1.5 mL polypropylene vials (Microsolv
Technology Corporation, Leland, NC).  Vials were fitted with pre-slit caps and used as supplied.
The fraction collector program was set to allow continuous collection of 2 min integrated
samples and was manually started after take-off.  Carousels were pre-loaded before flight and
then manually switched out as they were filled.  The vials were unloaded from the carousels at
the end of each flight, recapped with solid caps (Microsolv Technology Corporation), packed in
coolers with ice packs, and shipped back to Colorado State University to be stored in a 2 °C cold
room until analysis began following completion of the study.
**2.3.  Off-line Analysis**
Each fraction collector vial was brought to room temperature and then analyzed for
levoglucosan as well as a suite of anions/organic acids and cations.  For each analysis, 300 μL
aliquots were transferred to polypropylene vials.  Only levoglucosan, water-soluble potassium,
and ammonium are discussed here and their analytical methods are explained below.
The levoglucosan analysis was performed on a Dionex DX-500 series ion chromatograph
with pulsed amperometric detection via an ED-50/ED-50A electrochemical cell.  This cell
includes two electrodes: a "standard" gold working electrode and pH-Ag/AgCl (silver/silver
chloride) reference electrode.  A sodium hydroxide gradient and a Dionex CarboPac PA-1
column (4 x 250 mm) were employed for the separation.  The complete run time was 59 min
with an injection volume of 100 μL.  More details on the method can be found in *Sullivan et al.*
[2011a,b, 2014, 2019].  Only levoglucosan could be detected in the WE-CAN samples (other less
abundant anhydrosugars were too low to detect in the PILS samples) and did not require
background correction.  The LOD for levoglucosan based on a sample collection time of 2 min
and air flowrate of 15 LPM was determined to be less than approximately 0.10 ng/m$^3$.
A Dionex ICS-3000 ion chromatograph was used to measure water-soluble potassium
and ammonium.  An eluent generator provided a concentration of 20 mM methanesulfonic acid
at a flowrate of 0.5 mL/min to perform the separation on a Dionex IonPac CS12A analytical
column (3 x150 mm).  The complete run time was 17 min with an injection volume of 190 μL.
A blank correction was necessary for both of these species unlike levoglucosan.  Therefore, their


concentrations were corrected by using the average of all background samples collected during a
specific flight. For water-soluble potassium and ammonium, the LOD was 1 ng/m³.
**2.4. Photoacoustic Aerosol Absorption Spectrometer**
A custom-built PAS was used to measure total aerosol absorption at 405 and 660 nm
[*Foster et al.*, 2019] every 1 s during WE-CAN. The PAS measures aerosol light absorption at
near-ambient conditions by heating particles using a controlled light source and detecting the
resulting soundwave. It can be subject to interference by gaseous absorbers and is sensitive to
variations in relative humidity, temperature, and pressure [*Arnott et al.*, 1999; *Langridge et al.*,
2013]. The PAS sampled from a Solid Diffuser Inlet (SDI) mounted on the front right side of the
NSF/NCAR C-130. Aerosol passed through a cyclone impactor before entering the PAS to
remove particles with aerodynamic diameters > 1 µm. The flowrate for the PAS was 4 LPM.
Upstream of the PAS was a denuder to remove $NO_x$ (nitrogen oxides) from the sample air as
well as a Perma Pure dryer to dry the aerosol to below 30% relative humidity. A filter was
periodically switched in-line before the PAS to remove particles and allow for a near real-time
measurement of the baseline stability. Additionally, the PAS switched between sampling with
and without a thermal denuder system in-line. Only the data from sampling without the thermal
denuder is presented here. The PAS BrC absorption at 405 nm (PAS total Abs 405 BrC) was
calculated using equation 9 from *Pokhrel et al.* [2017]. This approach assumes the absorption
determined by the PAS at 660 nm was equivalent to BC absorption and the BC aerosol AAE was
1.
**2.5. Ultra-High-Sensitivity Aerosol Spectrometer**
One second particle number concentrations were measured using a rack-mounted
UHSAS (Ultra-High-Sensitivity Aerosol Spectrometer). The rack-mounted UHSAS switched
between sampling from the SDI inlet and a CVI (counter-flow virtual impactor) when sampling
out of and in-cloud, respectively. We only present data for sampling out of clouds. The rack-
mounted UHSAS was operated so that the flow could be manually lowered by the in-flight
operator when the NSF/NCAR C-130 flew through smoke plumes to allow the UHSAS to stay
within its optimum concentration measurement range. The particle size bins for the UHSAS
were calibrated using ammonium sulfate rather than traditional PSL (polystyrene latex) spheres.
Particle mass concentrations for $PM_1$ were calculated by applying these size bins and then
multiplying by 1.4 g/cm³ to account for particle density.
**2.6. Other Measurements**
In the following analysis, we focus on characterizing the BrC absorption in smoke from
wildfires in the western U.S. sampled during WE-CAN. Other airborne measurements used in
this analysis include meteorological data and coordinates provided by the Research Aviation
Facility (RAF) as part of the C-130 instrumentation package
(https://data.eol.ucar.edu/project/WE-CAN) and one Hz carbon monoxide (CO) determined by a
vacuum UV (ultraviolet) resonance fluorescence method [*Gerbig et al.*, 1999]. All data
presented in our analysis are reported at 1 atm and 273 K. Data from all species have been
averaged to match the 2 min collection time of the PILS-fraction collector system.
**2.7. Mie Calculation**





Mie calculations were performed by putting the complex refractive index (m = n+ ik) into
Mie code to obtain the absorption efficiency (Q) and then further calculate the absorption
coefficient using Eq. 1 [*Liu et al.*, 2013]. The real part of the refractive index (n) was set to be
1.55 and the imaginary part was calculated using Eq. 2 [*Liu et al.*, 2013].

$\beta\left(\lambda, D_p\right) = \frac{3}{2} \cdot \frac{Q \cdot WSOC}{D_p \cdot \rho}$         (Eq. 1)

$k = \frac{\rho \lambda \cdot H_2O\_\beta(\lambda)}{4\pi \cdot WSOC}$         (Eq. 2)

In Eq. 1 and 2, $\lambda$ is the wavelength, $D_p$ is the diameter of the particle, $\beta$ is the absorption
coefficient (referred to as the Mie calculated water-soluble absorption hereinafter), $Q$ is the
absorption efficiency, $WSOC$ is the water-soluble organic carbon mass concentration measured
by the PILS, and $H_2O\_\beta(\lambda)$ is the water-soluble light absorption coefficient measured by the
PILS. The particle density ($\rho$) was assumed to be 1.4 g/cm$^3$. The plume averaged particle size
distribution (measured by the UHSAS) was used in the calculation. The Mie calculated water-
soluble absorption was determined for each size bin in order to obtain the most accurate results.
Mie calculated total absorption was further calculated by multiplying the Mie calculated water-
soluble absorption by (UHSAS mass)/(WSOC*1.6), where the factor of 1.6 was to convert
WSOC to WSOM (water-soluble organic matter) [*Duarte et al.*, 2019; *Yttri et al.*, 2007].

**3. Results and Discussion**
**3.1. Overview**
Most previous studies employing a LWCC to determine water-soluble absorption,
examine the absorption at 365 nm (e.g., [*Hecobian et al.*, 2010; *Zhang et al.*, 2011, 2013]. But
here in order to explore the relationship between the water-soluble and total absorption
determined by the PILS and PAS, respectively, we focus on the absorption at 405 nm determined
by the LWCC. Using as examples Flight RF02, which sampled the Carr Fire smoke plume, and
Flight RF11, which sampled the Goldstone, Rabbit Foot, Beaver Creek, and Shellrock Fire
smoke plumes, Figures 2a and b show the relationship of the PILS water-soluble Abs 405 vs.
Abs 365. Absorption values at these two wavelengths are correlated (R$^2$ values from 0.70 to
1.00 based on all individual WE-CAN Flights), but the absorption measured at 405 nm was
about half of that observed at 365 nm (slope average 0.45 and range from 0.39 to 0.52 across all
individual WE-CAN flights).
Figure 3 shows example time series for WSOC, PILS water-soluble Abs 405, and PAS
total Abs 405 BrC from the same two flights as above. It was observed that all three parameters
tracked each other in and out of the smoke plumes. During WE-CAN, the average value ±
standard deviation for WSOC, PILS water-soluble Abs 405, and PAS total Abs 405 BrC were
13.35 ± 16.80 µg C/m$^3$, 6.06 ± 6.88 Mm$^{-1}$, and 22.02 ± 49.16 Mm$^{-1}$, respectively. The water-
soluble absorption determined by the PILS was lower than the total absorption determined by the
PAS. This pattern was consistently observed for all the wildfires sampled throughout WE-CAN.





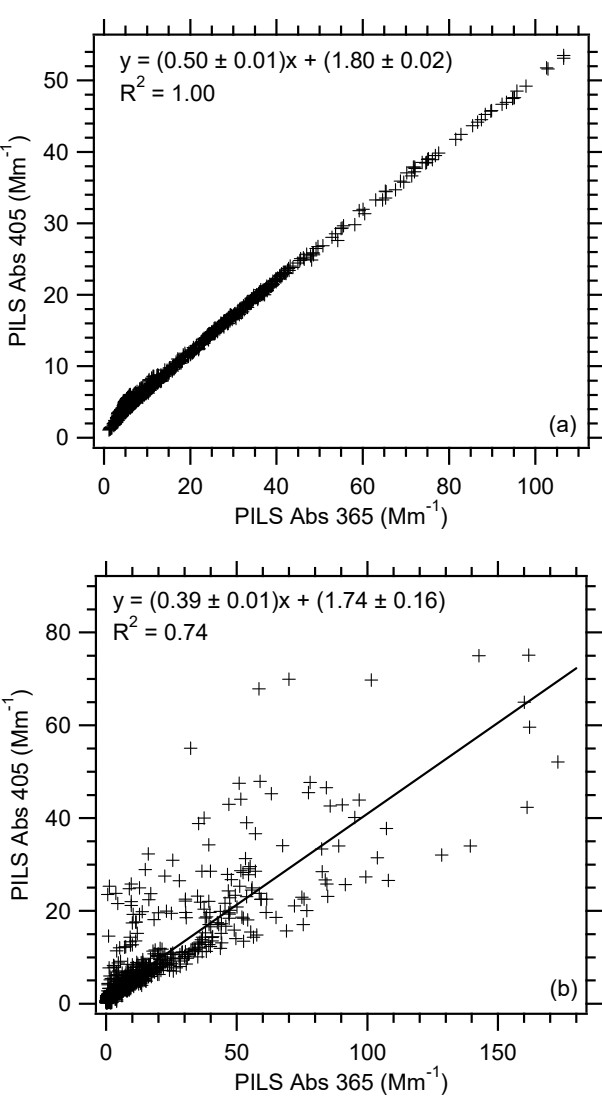

Figure 2. Correlation of PILS water-soluble Abs 405 vs. PILS water-soluble Abs 365 for WE-CAN (a) Flight RF02 and (b) Flight RF11. Uncertainties with the least square regressions are one standard deviation.

**3.2. Relationship between Total and Water-Soluble BrC Absorption**
To further explore the relationship between total and water-soluble BrC absorption, we
examine the relationship between PAS total Abs 405 BrC and UHSAS mass for Flights RF02
and RF11. There is a strong correlation between PAS total Abs 405 BrC and UHSAS mass
(Figure 4). Therefore, the PILS water-soluble Abs 405 can be corrected for the non-water-



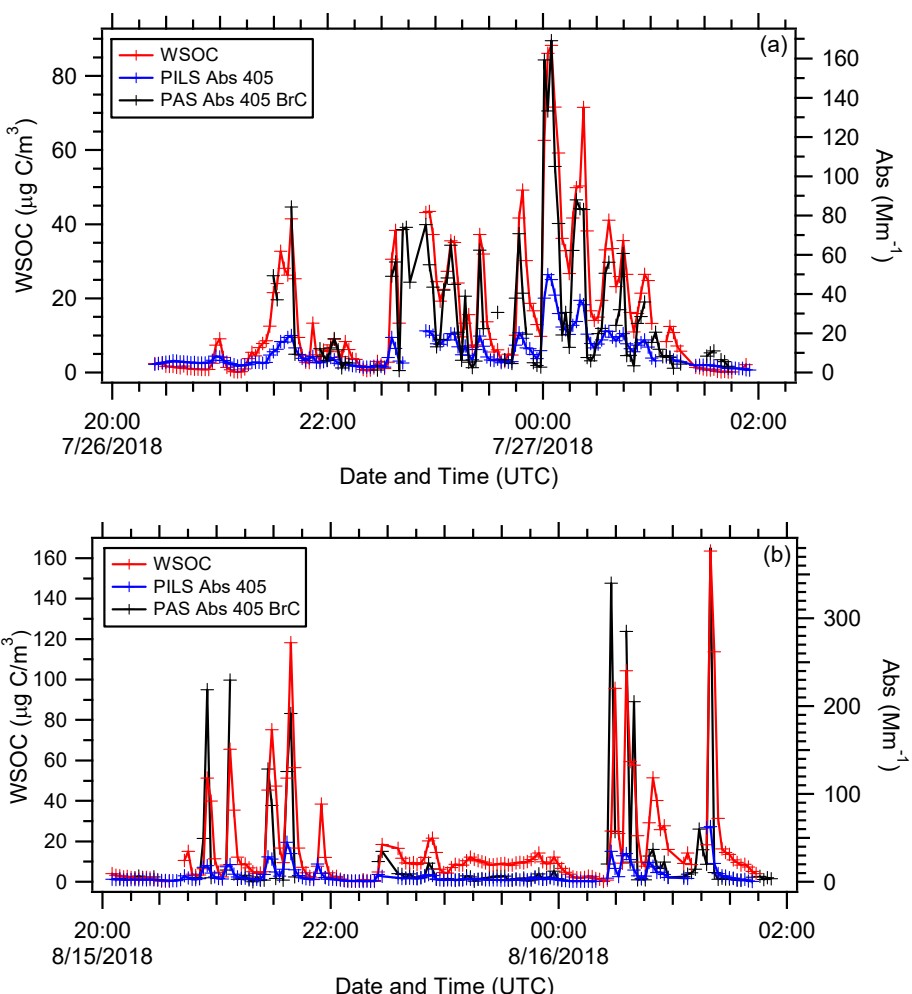

Figure 3. Time series of WSOC, PILS water-soluble Abs 405, and PAS total Abs 405 BrC for WE-CAN (a) Flight RF02 and (b) Flight RF11.

soluble fraction of the aerosol using the UHSAS mass. This was achieved by multiplying the
PILS water-soluble Abs 405 by 1/((WSOC*1.6)/(UHSAS mass)). This approach assumes the
water-soluble and non-water-soluble components of OC are the same.
Correcting the PILS water-soluble Abs 405 by the UHSAS mass showed good closure
with the PAS total Abs 405 BrC, but with a factor of ~1.5 to 2 difference between the PILS
water-soluble Abs 405 corrected and PAS total Abs 405 BrC (Figures 4c and d). This is also
similar to results obtained from the sampling of wildfire smoke during the FIREX (Fire Influence
on Regional and Global Environments Experiment) Campaign, where there was a ratio of 3.2
between PAS Abs 405 BrC and water-soluble Abs 405 determined from off-line LWCC analysis



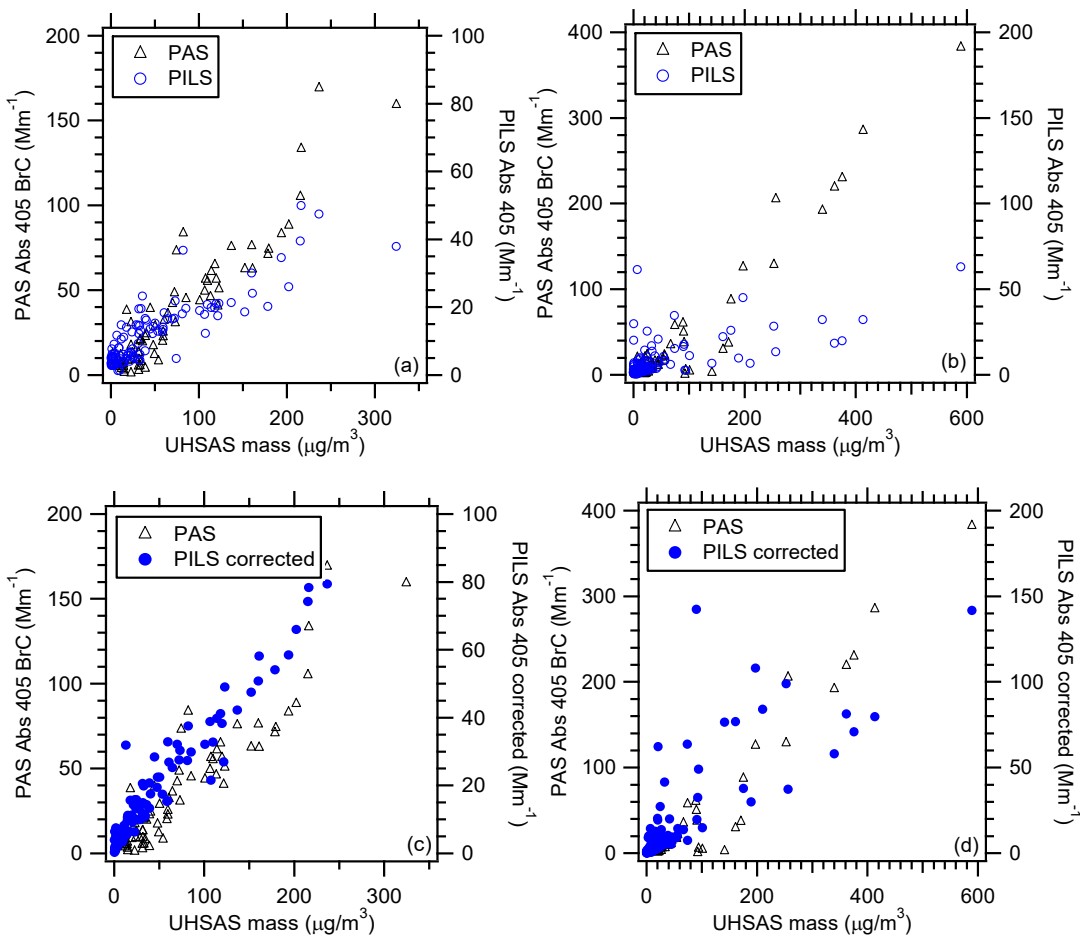

Figure 4. Correlation of PAS total Abs 405 BrC and PILS water-soluble Abs 405 vs. UHSAS mass
for WE-CAN (a) Flight RF02 and (b) Flight RF11. Correlation of PAS total Abs 405 BrC and PILS
water-soluble Abs 405 corrected for the non-water-soluble fraction of the aerosol using the UHSAS
mass for WE-CAN (c) Flight RF02 and (d) Flight RF11.




of filter samples [*Zeng et al.*, 2020]. This factor difference in both the WE-CAN and FIREX
data is likely due to the differences in particle vs. bulk solution absorption measured by the PAS
vs. LWCC (using PILS or filter samples), respectively, and can be explained by Mie Theory.

We used Mie Theory to calculate the water-soluble and total particle Abs 405 (see section

2.7) through each plume transect for RF02 and RF11. As shown in Figures 5a and b, we found a
slope of 1.7 to 1.8 for Mie calculated water-soluble Abs 405 to PILS Abs 405 and 3 to 4 for Mie
calculated total Abs 405 to PILS Abs 405. This is similar to results presented in *Liu et al.* [2013]
and based on off-line LWCC analysis of filter samples collected at 3 sites in Georgia. In that
work, a ratio of 2 for Mie calculated water-soluble Abs 365 to measured water-soluble Abs 365



and a ratio of 3.6 for Mie calculated total Abs 365 to measured water-soluble Abs 365 were
observed. In *Zeng et al.* [2022], Mie Theory was used to calculate the factor to convert solution
to particle light absorption (i.e., ratio of Mie calculated to measured water-soluble absorption) as
a function of wavelength for the FIREX data. At 405 nm a factor of ~1.7 was determined,
similar to what was determined from the WE-CAN data.
As a further check on the calculations performed here, the PAS Abs 405 BrC was
compared to the Mie calculated total Abs 405. Slopes ranged from 1.04 to 1.08 (Figures 5c and
d). This suggested our approach for correcting the PILS water-soluble Abs 405 for the non-
water-soluble fraction as well as to calculate the BrC absorption from the PAS Abs 405 data
were valid.
Overall, during WE-CAN ~45% (ranging from 31% to 65%) of the BrC absorption at
Abs 405 was due to water-soluble species. This is similar to what was observed from off-line
LWCC analysis of water and methanol extracts from filter samples collected during sampling of
biomass burning plumes as part of the DC3 (Deep Convective Clouds and Chemistry),
SEAC4RS (Studies of Emissions, Atmospheric Composition, Clouds and Climate Coupling by
Regional Surveys), and FIREX aircraft campaigns [*Forrister et al.*, 2015; *Liu et al.*, 2015; *Zeng
et al.*, 2022].
**3.3. BrC Absorption, CO, WSOC, and Levoglucosan**
Using data from all WE-CAN flights, Figure 6 shows that the PAS total Abs 405 BrC and
PILS water-soluble Abs 405 are correlated with CO ($R^2$ value for PAS = 0.76 and PILS = 0.55).
This further illustrates the importance of biomass burning as a source of BrC absorption (e.g.,
[*Andreae and Gelencsér*, 2006; *Chakrabarty et al.*, 2010; *Duarte et al.*, 2005; *Hecobian et al.*,
2010; *Hoffer et al.*, 2006; *Lack et al.*, 2012; *Lukács et al.*, 2007]).
Figure 7 shows that there is a correlation between BrC absorption and WSOC ($R^2$ value
for PAS = 0.42 and PILS = 0.60). This is not surprising given that the two main sources of
WSOC are typically biomass burning and secondary organic aerosol (SOA) [*Sullivan et al.*,
2006]. A number of previous studies where the source of WSOC and Abs 365 was one or both
of these have observed a similar correlation (e.g., [*Hecobian et al.*, 2010; *Liu et al.*, 2015; *Zhang
et al.*, 2013]). Additionally, analysis of cloud water samples impacted by biomass burning has
shown that nitrophenols and nitrocatechol are major contributors to the light absorption between
300 and 400 nm [*Desyaterik et al.*, 2013].
BrC absorption has a similar relationship with CO and WSOC as the biomass burning
marker levoglucosan [*Simoneit et al.*, 1999], but there are additional features (Figures 8a and b).
There is some variability in the ratio of levoglucosan to the PAS total Abs 405 BrC and PILS
water-soluble Abs 405 between wildfires, and this leads to two branches (Branch 1 and Branch
2). This was also observed for levoglucosan vs. WSOC (not shown). While there is no overall
correlation of levoglucosan vs. BrC absorption across all flights, there are correlations between
these two species on an individual flight basis (e.g., $R^2$ value for Flight RF02 = 0.76 and Flight
RF11 = 0.60, not shown). When data from all flights are colored by the water-soluble potassium
concentration (Figures 8c and d), we observe that Branch 1, which had the highest levoglucosan
concentrations, also has the highest water-soluble potassium concentrations (> 0.5 µg/m$^3$).
Levoglucosan and BrC absorption are much more highly correlated in Branch 1, than in Branch
2 for both the PILS ($R^2$ values Branch 1 = 0.76 and Branch 2 = 0.35) and PAS ($R^2$ values Branch
1 = 0.60 and Branch 2 = 0.22) BrC absorption. To further examine this, the times series of PILS



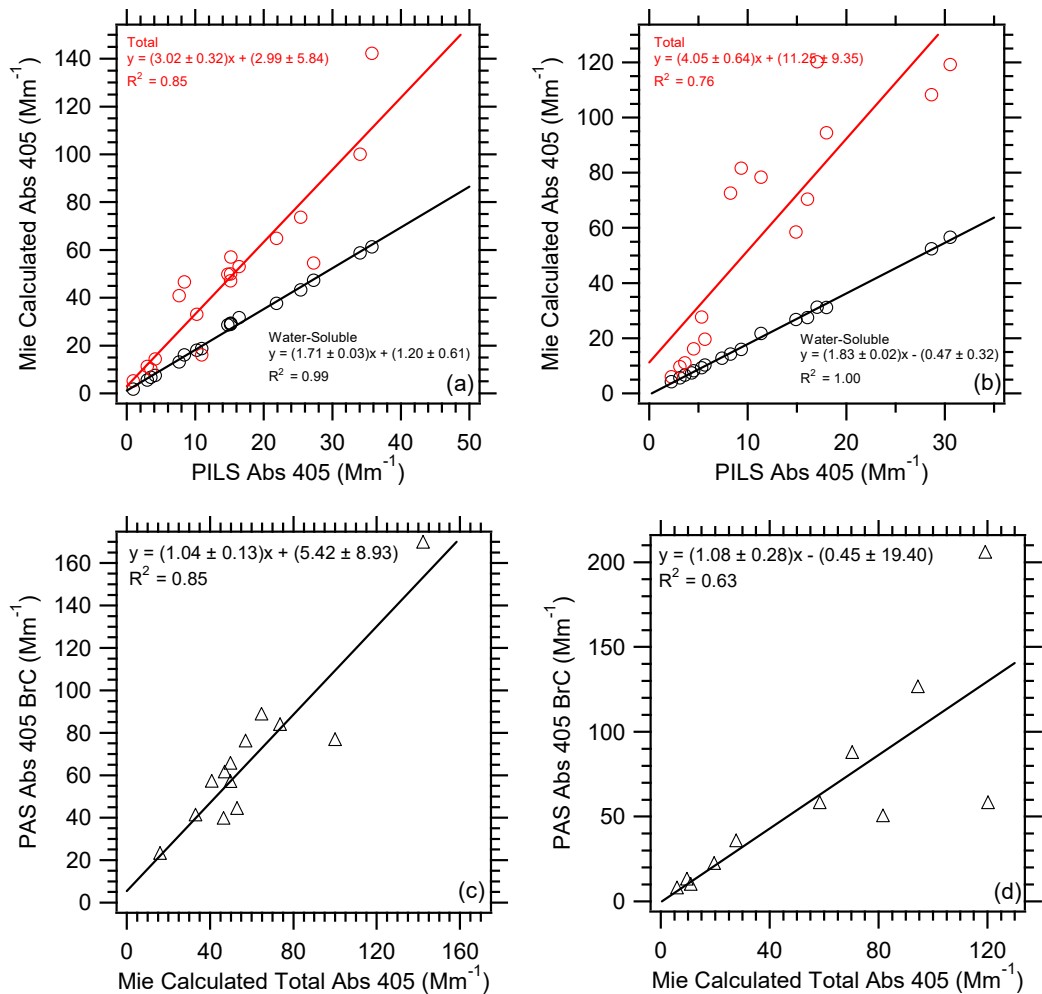

Figure 5. Correlation of Mie calculated water-soluble and total Abs 405 vs. PILS water-soluble Abs 405 for WE-CAN (a) Flight RF02 and (b) Flight RF11. Correlation of PAS total Abs 405 BrC and Mie calculated total Abs 405 for WE-CAN (c) Flight RF02 and (d) Flight RF11. Uncertainties with the least square regressions are one standard deviation.




water-soluble Abs 405, levoglucosan, potassium, and ammonium for Flights RF02 and RF11 are
shown in Figure 9. Smoke impacted samples in Flight RF02 had higher concentrations of
levoglucosan and water-soluble potassium and contributed to Branch 1. The data from Flight
RF11 contributed to Branch 2. In addition, elevated water-soluble potassium was observed in
many of the plume intercepts during Flight RF02. But more elevated ammonium was observed
for Flight RF11, which became even more prominent in smoke intercepts after 00:00 UTC, while
water-soluble potassium was relatively less abundant. Water-soluble potassium is a known



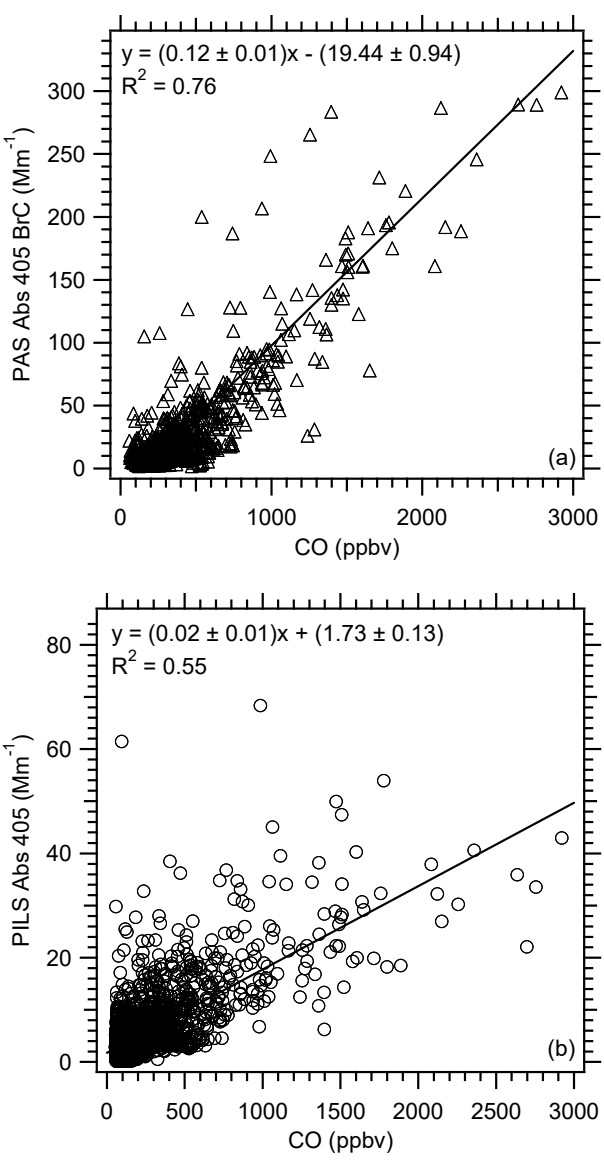

Figure 6. Correlation of (a) PAS total Abs 405 BrC and (b) PILS water-soluble Abs 405 vs. CO for all WE-CAN flights used in this analysis. Uncertainties with the least square regressions are one standard deviation.




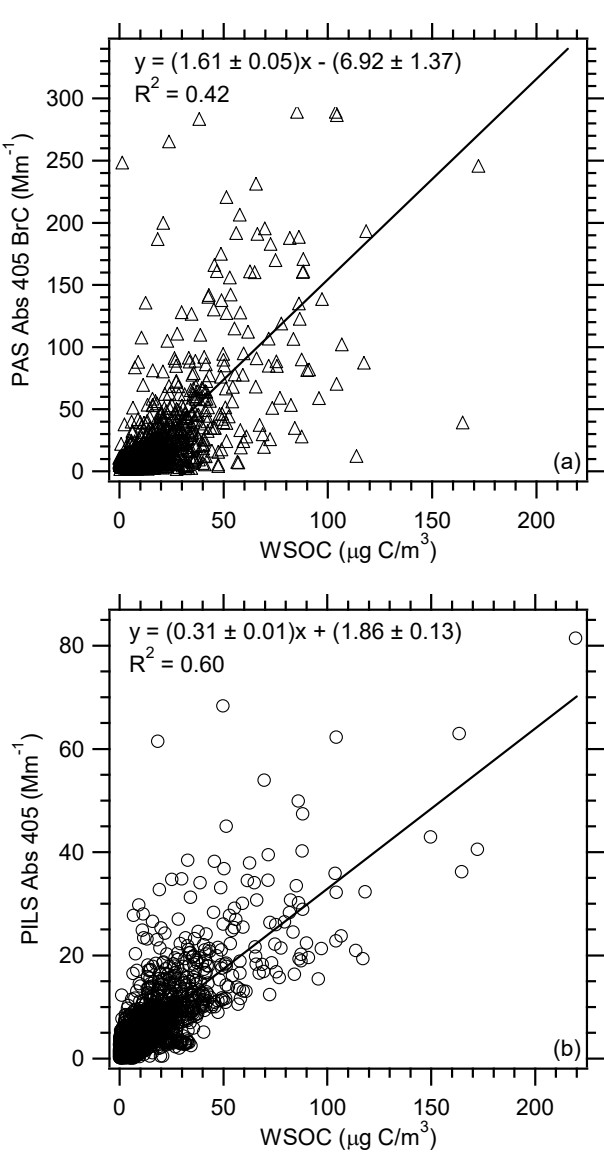

Figure 7. Correlation of (a) PAS total Abs 405 BrC and (b) PILS water-soluble Abs 405 vs. WSOC for all WE-CAN flights used in this analysis. Uncertainties with the least square regressions are one standard deviation.




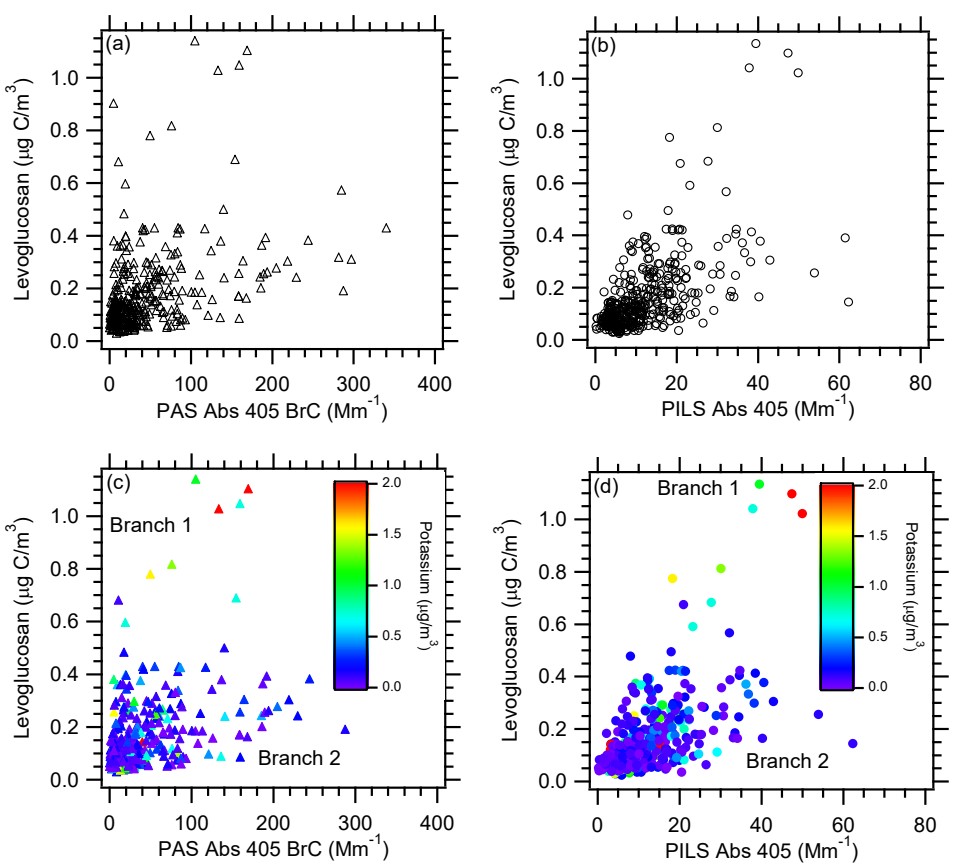

Figure 8. Correlations of levoglucosan on a carbon mass basis vs. (a) PAS total Abs 405 BrC
and (b) PILS water-soluble Abs 405 for all WE-CAN flights used in this analysis. Plots (c)
and (d) are the same as plots (a) and (b), but with the data colored by the PILS water-soluble
potassium concentrations. Branch 1 represents data with water-soluble potassium
concentrations > 0.5 µg/m³ and Branch 2 < 0.5 µg/m³. In plot (c), the equation for the fit and
$R^2$ value for Branch 1 are y = (0.006 ± 0.001)x + (0.027 ± 0.049), $R^2$ = 0.60 and for Branch 2
y = (0.001 ± 0.001)x + (0.118 ± 0.006), $R^2$ = 0.22, respectively. In plot (d), the equation for
the fit and $R^2$ value for Branch 1 are y = (0.024 ± 0.002)x - (0.081 ± 0.038), $R^2$ = 0.76 and for
Branch 2 y = (0.006 ± 0.001)x + (0.073 ± 0.007), $R^2$ = 0.35, respectively. Uncertainties with
the least square regressions are one standard deviation.

inorganic marker for biomass burning, although it is not as specific of a marker as levoglucosan
as there are additional possible sources for water-soluble potassium [*Schauer et al.*, 2001] and
water-soluble potassium is predominately emitted during only the flaming phase of a fire [*Lee et
al.*, 2010]. These results from WE-CAN suggest there may be a relationship between
levoglucosan and water-soluble potassium in wildfire emissions that has not been observed in
other types of burning [*Sullivan et al.*, 2014, 2019].



Figure 9. Time series from top to bottom of PILS water-soluble Abs 405, PILS levoglucosan, and PILS ammonium and water-soluble potassium for WE-CAN (a) Flight RF02 and (b) Flight RF11.




### 3.4. Evolution of BrC Absorption with Plume Age and Fire Dynamics

The time since emission (i.e., the smoke age) was estimated for all possible wildfire plumes as the distance the plume was sampled from the source divided by the average wind speed at that particular sampling altitude. Only PILS-fraction collector samples that directly overlapped with a CO plume penetration are considered. To account for dilution, we normalized the BrC absorption to 3 different species. We examine the ratio of BrC absorption to WSOC, $\Delta CO$ (assuming a CO background of 100 ppbv), and levoglucosan.

Figure 10a presents the ratio of PAS total Abs 405 BrC to WSOC and PILS water-soluble Abs 405 to WSOC, Figure 10b presents the ratio of PAS total Abs 405 BrC to $\Delta CO$ and PILS water-soluble Abs 405 to $\Delta CO$, and Figure 10c presents the ratio of PAS total Abs 405 BrC to levoglucosan and PILS water-soluble Abs 405 to levoglucosan as a function of time since emission. Figures S1-S3 show these 3 ratios for each smoke plume individually. If WSOC was lost with age due to evaporation of more volatile components or SOA formation were occurring with time since emission, CO would be expected to be more stable. It appears, however, that a similar pattern, perhaps with a bit more scatter for Abs 405 to WSOC, is observed for all of these ratios. Within a particular wildfire, there is no clear evidence that the PILS water-soluble BrC absorption is affected by smoke age up to 9 h. For the PAS total BrC absorption, especially for the ratio to $\Delta CO$, there appears to be a possible decrease in the ratio in the first 2 h, suggesting a need to further explore changes in total BrC absorption near the source region.

A number of laboratory studies suggest the initial stages of photochemical aging increases light absorption (i.e., photoenhancement). This is then followed by a decrease in light absorption (i.e., photobleaching) [*Hems and Abbatt*, 2018; *Saleh et al.*, 2013; *Sumlin et al.*, 2017; *Zhao et al.*, 2015; *Zhong and Jang*, 2014]. However, it is challenging to directly compare this laboratory data to the ambient data collected during WE-CAN. But analysis of laboratory and ambient biomass burning samples by *Wong et al.* [2019], found low molecular weight (< 400 Da) BrC undergoes rapid photobleaching on timescales of a few h, but high molecular weight (> 400 Da) BrC was stable for up to a few days. This suggest that the BrC sampled during WE-CAN could be composed mainly of high molecular weight species.

In addition, to investigate these ratios as a function of time since emission, the WE-CAN data had to be integrated across a smoke plume in order to incorporate the PILS-fraction collector measurements. Of course, a smoke plume itself was dynamic with concentrations being highest in the middle of the plume and more dilute on the edges. It is possible the averaging could contribute to the observed pattern of BrC absorption not changing with age. *Forrister et al.* [2015], who used plume transect averages of SEAC4RS data, reported a decrease in the total Abs 365/$\Delta CO$ from ~0.13 to 0.07 Mm$^{-1}$/ppbv in 5 h for smoke from the Rim Fire. Observations of smoke during FIREX, by contrast, indicated no clear trend with plume age [*Zeng et al.*, 2022] in a dataset where the majority of plume ages were less than 10 h. These varying results also suggest that other factors that contribute to changes in BrC absorption over time may still need to be explored.

In order to investigate the possible influence of fire dynamics on BrC absorption, the modified combustion efficiency (MCE) was calculated as the change in carbon dioxide divided by the sum of the change in carbon monoxide and carbon dioxide ($\Delta CO_2/(\Delta CO +\Delta CO_2)$) on a molar basis [*Ward and Radke*, 1993]. A higher MCE value indicates a more intense or extended flaming phase as opposed to a smoldering phase. Within a particular wildfire there appeared to be no clear dependence of the ratio of BrC absorption to WSOC, $\Delta CO$, or levoglucosan on MCE



Figure 10. (a) Abs 405/WSOC, (b) Abs 405/ΔCO, and (c) Abs 405/levoglucosan as a function of time since emission for all WE-CAN flights with the data segregated by flight. In each plot the PAS total Abs 405 BrC is on top and the PILS water-soluble Abs 405 on the bottom.



(Figure 11 and Figures S4-S6), except that an overall lower Abs 405/levoglucosan ratio was
observed for the wildfires with higher MCE values (i.e., Flight RF02).  This further supports the
relationship between the highest potassium concentrations and the levoglucosan vs. Abs 405
correlation (Figures 8c and 8d) previously discussed as potassium is predominately emitted from
the flaming phase of a fire [*Echalar et al.*, 1995; *Lee et al.*, 2010; *Ward et al.*, 1991].
**4. Summary**
A PILS-LWCC-TOC and PAS were deployed on the NSF/NCAR C-130 research aircraft
during WE-CAN to examine aerosol absorption in wildfire smoke in the western U.S.  This was
the first deployment of the PILS-LWCC-TOC on a research aircraft.  The PILS allowed for a 16
s integrated measurement of the water-soluble BrC absorption and 4 s integrated measurement of
WSOC.  The data from the PILS and PAS were combined to investigate the water-soluble vs.
total BrC absorption at 405 nm in the 20 wildfires sampled during WE-CAN.  We show the
following:
1.  WSOC, PILS water-soluble Abs 405, and PAS total Abs 405 BrC tracked each other in
and out of the smoke plumes.  BrC absorption was correlated with CO ($R^2$ value for PAS
= 0.76 and PILS = 0.55) and WSOC ($R^2$ value for PAS = 0.42 and PILS = 0.60) during
the entire study, illustrating the importance of biomass burning as a source of BrC
absorption.  A similar pattern was observed for levoglucosan, but with two data branches.
Levoglucosan and BrC absorption were correlated ($R^2$ values for PAS = 0.60 and PILS =
0.76) in the first data branch and this subset of data was also characterized by the highest
observed water-soluble potassium concentrations (> 0.5 µg/m$^3$).  This suggests there may
be a relationship between levoglucosan and water-soluble potassium in wildfire
emissions that has not generally been observed in other types of burning.
2.  Using the calculated UHSAS mass, the PILS water-soluble Abs 405 can be corrected to
also account for the non-water-soluble fraction of the aerosol.  The corrected PILS water-
soluble Abs 405 showed good closure with the PAS total Abs 405 BrC, but with a factor
of ~1.5 to 2 difference.  This difference can be explained by particle vs. bulk solution
absorption measured by the PAS vs. PILS, respectively, as shown by Mie Theory
calculations.  During WE-CAN, ~45% of the BrC absorption was due to water-soluble
species.
3.  The ratio of water-soluble BrC absorption to WSOC, ΔCO, or levoglucosan showed no
clear dependence on fire dynamics or the time since emission up to 9 h.  The total BrC
absorption did show a slight decrease in the first 2 h, suggesting a need to further explore
near source evolution.



Figure 11. (a) Abs 405/WSOC, (b) Abs 405/ΔCO, and (c) Abs 405/levoglucosan as a function of modified combustion efficiency for all WE-CAN flights with the data segregated by flight. In each plot the PAS total Abs 405 BrC is on top and the PILS water-soluble Abs 405 on the bottom.





**Data Availability**

The WE-CAN data is provided by NCAR/EOL under sponsorship of the National Science
Foundation and is available at http://data.eol.ucar.edu/master_lists/generated/we-can/. The DOI
for each data set used in this work are:

PILS1: https://doi.org/10.26023/9H07-MD9K-430D and https://doi.org/10.26023/CRHY-NDT9-
C30V
PILS2: https://doi.org/10.26023/7TAN-TZMD-680Y
PAS: https://doi.org/10.26023/K8P0-X4T3-TN06
UHSAS: https://doi.org/10.26023/BZ4F-EAC4-290W
CO: https://doi.org/10.26023/NNYM-Z18J-PX0Q
Meteorological Data and Coordinates: https://doi.org/10.26023/G766-BS71-9V03

**Author Contributions**

APS, SMM, DWT, EVF, JLC designed the project. APS wrote the paper. APS, RPP, YS,
SMM, DWT, TC, JL, and EVF collected and analyzed data. All authors reviewed and provided
comments for the paper.

**Conflict of Interest**

The authors declare that they have no conflict of interest.

**Acknowledgements**

We wish to thank RAF personnel for their many contributions supporting the field deployment.
We also thank R.J. Weber for generously providing some of the parts used in the PILS racks.

**Financial Support**

This work was supported by the National Science Foundation under AGS-1650786.



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
