# Peer review of "Examination of Brown Carbon Absorption from Wildfires in the Western U.S. During the"

_Atmospheric Chemistry and Physics, 2022_

## Author Comment (AC1)

Reviewer 1
Line 265-268: Might want to discuss issues with ignoring lensing of BC, see Pokhrel. What about the other assumption of no BrC at 660 nm? Does this matter in the following analysis?

The text had been updated to include a discussion about ignoring lensing of BC as part of the approach used to calculated BrC from the PAS measurements. Lines 266-271 now read as "The PAS BrC absorption at 405 nm (PAS total Abs 405 BrC) was calculated using equation 9 from *Pokhrel et al.* [2017]. This approach assumes the absorption determined by the PAS at 660 nm was equivalent to BC absorption, the BC aerosol AAE was 1, and the absorption enhancement from lensing was constant at all wavelengths. Previous work using this approach in smoke from controlled laboratory burns found lensing could contribute a maximum of 30% of total absorption, but typical contributes much less."

A separate paper in preparation by co-author Shen using the WE-CAN data set is looking at the absorption at 660 nm and what role BrC may play.

Give the particle size range measured by the UHSAS and what size range was used to calculate overall particle mass concentration.

The UHSAS measures particles in the 0.06-1 µm range. The text has been updated to indicate this and Lines 280-284 now read as "The UHSAS measures particles in the 0.06-1 µm range. The particle size bins for the UHSAS were calibrated using ammonium sulfate rather than traditional PSL (polystyrene latex) spheres. Particle mass concentrations for $PM_1$ were calculated by summing all size bins and then multiplying by 1.4 $g/cm^3$ to account for particle density."

What is the cause of the large difference in correlations in Figs 2a vs 2b? The poor correlation in 2b is somewhat surprising and suggests the AAE was highly variable. Is this because 2a is data from one fire and 2b is from several different fires? This difference in correlation between the two flight is seen throughout the analysis. This deserves more investigation. If it is caused by high variability due to smoke from different fires for RF11, then maybe a more uniform plume should be used, although this flight does provide a contrast.

The difference in relationship seen for Abs 405 vs. Abs 365 for Flights RF02 and RF11 is due to RF11 sampling narrower plumes than RF02, which sampled much broader plumes. We had initially included the range of $R^2$ values seen for this relationship from all flights to indicate we selected these 2 flights to cover the range. But to more clearly indicate this we updated the text and Lines 326-333 now read as "Absorption values at these two wavelengths are correlated ($R^2$ values from 0.70 to 1.00 based on all individual WE-CAN Flights), but the absorption measured at 405 nm was about half of that observed at 365 nm (slope average 0.45 and range from 0.39 to 0.52 across all individual WE-CAN flights). We selected these two flights to cover the range in the relationships observed during WE-CAN. The lower correlation for Flight RF11 is likely due to the narrower plumes being sampled compared to the broader plumes observed in Flight RF02. This difference related to narrow vs. broad plumes was observed throughout the various wildfires sampled during WE-CAN."

Slopes could be included in Fig 4 plots, which is the MCE.

Equations and $R^2$ values for least square regression fits for each plot in Figure 4 have been added to the caption. The caption now reads as "Figure 4. Correlation of PAS total Abs 405 BrC and PILS water-soluble Abs 405 vs. UHSAS mass for WE-CAN (a) Flight RF02 and (b) Flight RF11. Correlation of PAS total Abs 405 BrC and PILS water-soluble Abs 405 corrected for the non-water-soluble fraction of the aerosol using the UHSAS mass for WE-CAN (c) Flight RF02 and (d) Flight RF11. In plot (a), the equations for the fit and $R^2$ value for PAS are y = (0.50 ± 0.02)x – (0.03 ± 2.29), $R^2$ = 0.87 and for PILS are y = (0.14 ± 0.01)x + (5.58 ± 0.65), $R^2$ = 0.75, respectively. In plot (b), the equations for the fit and $R^2$ value for PAS are y = (0.62 ± 0.03)x – (6.09 ± 3.62), $R^2$ = 0.76 and for PILS are y = (0.08 ± 0.01)x + (3.58 ± 0.66), $R^2$ = 0.43, respectively. In plot (c), the equation for the fit and $R^2$ value for PILS are y = (0.32 ± 0.01)x + (3.60 ± 0.68), $R^2$ = 0.94, respectively. In plot (d), the equation for the fit and $R^2$ value for PILS are y = (0.24 ± 0.01)x + (2.93 ± 1.35), $R^2$ = 0.65, respectively. Uncertainties with the least square regressions are one standard deviation."

Why not write equation in line 348 as: UHSAS/(1.6 WSOC)?

The equation was updated as suggested and Lines 353-356 now read as "Therefore, the PILS water-soluble Abs 405 can be corrected for the non-water- soluble fraction of the aerosol using the UHSAS mass. This was achieved by multiplying the PILS water-soluble Abs 405 by 1/((WSOC*1.6)/(UHSAS mass)) or (UHSAS mass)/(WSOC*1.6)."

Line 349, what properties of WSOC and WIOC are assumed to be same? It appears the assumption is that the MCE is assumed the same for both, and that the total mass = total OA mass.

To better clarify, Lines 356-357 now read as "This approach assumes the characteristics of the non-water-soluble components of OC are identical to that of the water-soluble components of OC."

Fig 5a and b are somewhat confusing. The x-axis is measured WS absorption and the y axis is the Mie calculate for WS and total. What data was used as input to the Mie calculation in each case, for water soluble was it the MAE for soluble species and for total the MAE total calculated previously from the WS species?

Section 2.7 provides details on the Mie calculations performed. The measured WSOC concentration and size distribution data were used in the calculation for the water-soluble absorption. Then the total absorption was calculated from WSOC by converting WSOC to WSOM and using the ratio to total mass. We updated the text to better direct readers to section 2.7 and Lines 367-369 now read as "We used Mie Theory to calculate the water-soluble and total particle Abs 405 (see section 2.7 for details on the equations and parameters used) through each plume transect for RF02 and RF11."

What would happen if the same calculations were repeated for different wavelengths?

We only performed the calculations at 405 nm due to the two wavelengths the PAS measured at and no other optical measurements being conducted during WE-CAN. But we expect the Mie factor to change some. As shown in Zeng et al., *Atmos Chem. Phys.*, 2022, the Mie factor would likely decrease from ~1.7 at 405 nm to ~1.3 at 700 nm.

Lines 474 to 484, on the discussion of the evolution of smoke plumes. The results of this paper seem to contradict a study already published (Palm et al) based on these same (WE-CAN) data and which there are common co-authors to this manuscript. Palm concludes that although changes in the OA may be occurring, there tends to be a balance, so parameters like OA mass and BrC relative to CO remain steady as the smoke plume evolves. The authors should contrast their findings to this paper, since it seems to be two different interpretations from the same study with common co-authors. Additionally, the Palm paper should be cited and findings discussed in this papers Introduction. (Palm, B., Q. Peng, C. D. Fredrickson, B. H. Lee, L. A. Garofalo, M. A. Pothier, S. M. Kreidenweis, D. K. Farmer, R. P. Pokhrel, Y. Shen, S. M. Murphy, W. Permar, L. Hu, T. L. Capos, S. R. Hall, K. Ullmann, X. Zhang, F. Flocke, E. V. Fischer, and J. A. Thornton (2020), Quantification of organic aerosol and brown carbon evolution in fresh wildfire plumes, P. Natl. Acad. Sci., 117(47), 29469-29477.)

We initially did not include the Palm et al. reference here as their analysis did not examine the water-soluble Abs 405 and our paper is focused on the comparison of water-soluble Abs 405 and total Abs 405. In addition, a recent analysis on BrC in wildfire plumes by Zeng et al. has shown that the dilution-drive evaporation suggested in Palm et al. likely plays a minor role. The text has been updated so Lines 501-510 now reads as "An analysis of WE-CAN data by *Palm et al.* [2020] looking at the evolution of organic aerosol and BrC suggested that although changes in organic aerosol were likely occurring, there was a balance between dilution-driven evaporation and subsequent formation resulting in little change over time. It is hard to compare these results to our analysis as the *Palm et al.* [2020] work chose to focus only on the total organic aerosol and total Abs 405 BrC and did not examine the WSOC or water-soluble Abs 405. When examining the ratio of WSOC to ΔCO as a function of time since emission (Figure S5) during WE-CAN, there was not clear evidence for formation or loss of WSOC being observed within a particular wildfire. But a recent analysis by *Zeng et al.* [2022] has shown in wildfire plumes that dilution-drive evaporation was likely playing a minor role compared to the effects of ozone on BrC."

In Fig 10, trends for different flights can't be discerned, maybe add regression lines for each flight? Also, do a regression for PAS total BrC vs dCO to support the claim that there is a consistent drop in the first 2 hours (eg, on what bases was this conclusion reached)?

To be able to better discern trends in the different flights, is the exact reason that we provided the PILS Abs 405 and PAS Abs 405 BrC data individually for each flight in the supporting information as Figures S1-S3 for each of the three ratios shown in Figure 10. Following the typical analysis by other authors, we are not providing regression lines. But the drop in the total PAS Abs 405 BrC/ΔCO can be seen in plot S2i for Flight RF15, which best covers the first 2 h of time since emission. The text has been updated to reflect all of this and Lines 462-476 now read as "Figure 10a presents the ratio of PAS total Abs 405 BrC to WSOC and PILS water-soluble Abs 405 to WSOC, Figure 10b presents the ratio of PAS total Abs 405 BrC to ΔCO and

PILS water-soluble Abs 405 to ΔCO, and Figure 10c presents the ratio of PAS total Abs 405 BrC to levoglucosan and PILS water-soluble Abs 405 to levoglucosan as a function of time since emission.  To better discern any trends, Figures S1-S3 show these 3 ratios for each smoke plume on an individual flight basis.  If WSOC was lost with age due to evaporation of more volatile components or SOA formation were occurring with time since emission, CO would be expected to be more stable.  It appears, however, that a similar pattern, perhaps with a bit more scatter for Abs 405 to WSOC, is observed for all of these ratios.  Within a particular wildfire, there is no clear evidence that the PILS water-soluble BrC absorption is affected by smoke age up to 9 h.  For the PAS total BrC absorption, especially for the ratio to ΔCO, there appears to be a possible decrease in the ratio in the first 2 h (see Figures S1i and S2i for Flight RF15 which best covered this period), suggesting a need to further explore changes in total BrC absorption near the source region."

Maybe look at change in WSOC relative to CO to check specifically for production or loss of WSOC with plume age?

We did examine the ratio of WSOC/ΔCO as a function of time since emission and we see little change in the ratio.  We initially did not include this plot as we considered it beyond the scope of the work presented here as this paper focuses on the Abs 405.  But we have now included it in the supporting information and a reference to the figures appears in Lines 506-508 as "When examining the ratio of WSOC to ΔCO as a function of time since emission (Figure S5) during WE-CAN, there was not clear evidence for formation or loss of WSOC being observed within a particular wildfire."

[Figure]

Figure S5.  PILS WSOC/ΔCO as a function of time since emission for all WE-CAN flights with the data segregated by flight.

Conclusion 2, last line, give wavelength for the ratio of WS BrC to total.

Conclusion 2 has been updated as suggested and Lines 543-549 now read as "Using the calculated UHSAS mass, the PILS water-soluble Abs 405 can be corrected to also account for the non-water-soluble fraction of the aerosol.  The corrected PILS water-soluble Abs 405 showed good closure with the PAS total Abs 405 BrC, but with a factor of ~1.5 to 2 difference.  This difference can be explained by particle vs. bulk solution absorption measured by the PAS vs. PILS, respectively, as shown by Mie Theory calculations.  During WE-CAN, ~45% of the BrC absorption at 405 nm was due to water-soluble species."

Reviewer 2
There are other studies that have not been mentioned that examined relationships between organics and biomass burning tracers (e.g., Lee et al., (2016) and Di Lorenzo et al., (2018)). The discussion in this work would be stronger if it related these observations to those in these previous works. It would also be interesting to know more about the relationship (or lack thereof) between ammonium and brown carbon and/or other biomass burning trackers. Di Lorenzo et al. (2018) saw a relationship between reduced nitrogen and brown carbon in aged samples.
The discussion of brown carbon absorption with plume age should include discussion of these results in the context of previously published results from the same aircraft campaign (Palm et al., 2020). The section should also discuss recent aircraft PiLS work that has examined similar trends (Washenfelder et al., 2022).

The Lee et al. and Di Lorenzo et al. references have been added to Section 3.3 and Lines 446-449 now read as "It has been observed in previous work looking at the size-resolved aerosol composition and single particle measurements from wildfire plumes that water-soluble potassium and levoglucosan appear in different sized particles than BrC and that there is non-uniform mixing of them [*Di Lorenzo et al.*, 2018; *Lee et al.*, 2016], which could also be a factor."

In our samples, even when more elevated ammonium as opposed to potassium was observed, there was no correlation between ammonium and BrC found.  The text has been updated to indicate this and Lines 435-440 now read as "In addition, elevated water-soluble potassium was observed in many of the plume intercepts during Flight RF02.  But more elevated ammonium was observed for Flight RF11, which became even more prominent in smoke intercepts after 00:00 UTC, while water-soluble potassium was relatively less abundant.  While ammonium was clearly more prominent, there was no correlation observed between ammonium and PAS total Abs 405 BrC or PILS water-soluble Abs 405 for the data contributing to Branch 2 (not shown)."

Section 3.4 on the evolution of BrC absorption with plume age has been updated to include both the Palm et al. and Washenfelder et al. references.  Lines 493-496 now read as "Observations of smoke during FIREX, by contrast, indicated no clear trend with plume age [*Washenfelder et al.*, 2022; *Zeng et al.*, 2022] in a dataset where the majority of plume ages were less than 10 h. These varying results also suggest that other factors that contribute to changes in BrC absorption over time may still need to be explored."  Lines 501-510 now read as "An analysis of WE-CAN data by *Palm et al.* [2020] looking at the evolution of organic aerosol and BrC suggested that although changes in organic aerosol were likely occurring, there was a balance between dilution-driven evaporation and subsequent formation resulting in little change over time.  It is hard to compare these results to our analysis as the *Palm et al.* [2020] work chose to focus only on the total organic aerosol and total Abs 405 BrC and did not examine the WSOC or water-soluble Abs 405.  When examining the ratio of WSOC to ΔCO as a function of time since emission (Figure S5) during WE-CAN, there was not clear evidence for formation or loss of WSOC being observed within a particular wildfire.  But a recent analysis by *Zeng et al.* [2022] has shown in wildfire plumes that dilution-drive evaporation was likely playing a minor role compared to the effects of ozone on BrC."

Line 178: I think both PiLS sampled from the same inlet—"each" implies they have their own separate inlets. It would improve clarity to say "both PILS systems sampled from…"

Each PILS did actually sample from their own inlet.  To better clarify this, Lines 178-180 have been updated to read as "Each PILS system sampled from their own Submicron Aerosol Inlet (SMAI) [*Craig et al.*, 2013a, 2013b, 2014; *Moharreri et al.*, 2014] mounted to the belly of the NSF/NCAR C-130."

Line 230: Other anhydrosugars being below detection is mentioned later on in the manuscript. Suggest giving the more general term at this point in the methods, before later saying that you focused on levoglucosan.

The text has been updated as suggested and Lines 229-230 now read as "Each fraction collector vial was brought to room temperature and then analyzed for anhydrosugars as well as cations."

Line 230: Suggest not mentioning anions/organic acids since none of the data or methods are presented.

The text has been updated as suggested and Lines 229-230 now read as "Each fraction collector vial was brought to room temperature and then analyzed for anhydrosugars as well as cations."

Line 246: What detector was used here? I assume conductivity.

Yes, a conductivity detector was used for the cation measurements.  Lines 243-244 now read as "A Dionex ICS-3000 ion chromatograph equipped with a conductivity detector was used to measure water-soluble potassium and ammonium."

Line 389: What fraction of the samples was not affected by biomass burning? That will affect the robustness of this claim. Somewhere in the discussion of tracers, it might be worth explicitly stating why you might expect to see differences in correlations with CO versus the other tracers that are discussed.

More than 75% of the WE-CAN data was collected in smoke.  Lines 393-398 have been updated to reflect this and now read as "Using data from all WE-CAN flights, Figure 6 shows that the PAS total Abs 405 BrC and PILS water-soluble Abs 405 are correlated with CO ($R^2$ value for PAS = 0.76 and PILS = 0.55).  This further illustrates the importance of biomass burning as a source of BrC absorption (e.g., [*Andreae and Gelencsér*, 2006; *Chakrabarty et al.*, 2010; *Duarte et al.*, 2005; *Hecobian et al.*, 2010; *Hoffer et al.*, 2006; *Lack et al.*, 2012; *Lukács et al.*, 2007]), especially since more than 75% of the WE-CAN data occurred in smoke."

We have also added additional text to Section 3.3 to mention why we expect to see differences in the correlations of Abs 405 with CO versus the other tracers as suggested.  Lines 440-453 now read as "Water-soluble potassium is a known inorganic marker for biomass burning, although it is not as specific of a marker as levoglucosan as there are additional possible sources for water-soluble potassium [*Schauer et al.*, 2001] and water-soluble potassium is predominately emitted during only the flaming phase of a fire [*Lee et al.*, 2010].  It is possible this difference in timing of emissions is what leads to the different relationship of Abs 405 with levoglucosan than was observed for CO and WSOC.  It has been observed in previous work looking at the size-resolved

aerosol composition and single particle measurements from wildfire plumes that water-soluble potassium and levoglucosan appear in different sized particles than BrC and that there is non-uniform mixing of them [*Di Lorenzo et al.*, 2018; *Lee et al.*, 2016], which could also be a factor. These results from WE-CAN are further suggesting there may be a relationship between levoglucosan and water-soluble potassium in wildfire emissions that has not been observed in other types of burning, such as prescribed burning, residential burning, or controlled laboratory burns [*Sullivan et al.*, 2014, 2019]."

Lines 397-399: This sentence seems out of place. It is not clear how it connects to the preceding discussion.

This sentence has been removed as suggested.

Figure 8: I think parts (a) and (b) of this figure could be removed.

Plots a and b have been removed from Figure 8. The original plots c and d have been relabeled as plots a and b. The new Figure 8 is shown below:

[Figure]

[Figure]

Figure 8. Correlations of levoglucosan on a carbon mass basis vs. (a) PAS total Abs 405 BrC and (b) PILS water-soluble Abs 405 for all WE-CAN flights used in this analysis with the data colored by the PILS water-soluble potassium concentrations. Branch 1 represents data with water-soluble potassium concentrations $> 0.5$ $\mu g/m^3$ and Branch 2 $< 0.5$ $\mu g/m^3$. In plot (a), the equation for the fit and $R^2$ value for Branch 1 are $y = (0.006 \pm 0.001)x + (0.027 \pm 0.049)$, $R^2 = 0.60$ and for Branch 2 $y = (0.001 \pm 0.001)x + (0.118 \pm 0.006)$, $R^2 = 0.22$, respectively. In plot (b), the equation for the fit and $R^2$ value for Branch 1 are $y = (0.024 \pm 0.002)x - (0.081 \pm 0.038)$, $R^2 = 0.76$ and for Branch 2 $y = (0.006 \pm 0.001)x + (0.073 \pm 0.007)$, $R^2 = 0.35$, respectively. Uncertainties with the least square regressions are one standard deviation.

Line 444: Could you be more specific about the other types of burning you mean here?

This sentence has been updated to be more specific about the types of burning as suggested and Lines 449-453 now read as "These results from WE-CAN are further suggesting there may be a relationship between levoglucosan and water-soluble potassium in wildfire emissions that has not been observed in other types of burning, such as prescribed burning, residential burning, or controlled laboratory burns [*Sullivan et al.*, 2014, 2019]."

Line 470: There is field data that describes this phenomenon as well that may be of interest for comparison (though the aging times may be too long to be relevant, Di Lorenzo et al. (2017)).

The text has been updated to include reference to this ambient data and Lines 481-485 now read as "But analysis of laboratory and ambient biomass burning samples found low molecular weight ($< 400$ Da) BrC undergoes rapid photobleaching on timescales of a few h, but high molecular weight ($> 400$ Da) BrC was stable for up to a few days [*Di Lorenzo et al.*, 2017; *Wong et al.*, 2019]. This suggests that the BrC sampled during WE-CAN could be composed mainly of high molecular weight species."

Reviewer 3

1.For Figure 2, it is unclear why PILS Abs 365 has excellent correlation with PILS Abs 405 for research flight 02, but not so great for research flight 11? Was that because RF11 samples a lot more background air? Some explanation on the difference between these two flights would be useful. Also, it would be important to comment on this aspect for other flights.

As previously mentioned, the difference in the relationships observed for Abs 405 vs. Abs 365 for Flight RF02 and RF11 is due to RF11 sampling narrower plumes and RF02 sampling broader plumes. The range of $R^2$ values seen for this relationship from all flights was noted to indicate we had selected these 2 flights to cover the range observed during WE-CAN. But we updated the text to more directly indicate this and Lines 326-333 now read as "Absorption values at these two wavelengths are correlated ($R^2$ values from 0.70 to 1.00 based on all individual WE-CAN Flights), but the absorption measured at 405 nm was about half of that observed at 365 nm (slope average 0.45 and range from 0.39 to 0.52 across all individual WE-CAN flights). We selected these two flights to cover the range in the relationships observed during WE-CAN. The lower correlation for Flight RF11 is likely due to the narrower plumes being sampled compared to the broader plumes observed in Flight RF02. This difference related to narrow vs. broad plumes was observed throughout the various wildfires sampled during WE-CAN.

2.I kept wondering if the conclusion of no significant photobleaching within the first 9h after emission, could be a wavelength-specific problem. Can the authors add PILS Abs 365 to Figure 11? So, it would be clear whether a similar conclusion can be reached for the absorption at 365nm or even shorter wavelength.

We do not believe that our results presented here from the WE-CAN Abs 405 data are wavelength specific. The studies from FIREX mentioned in the text (*Washenfelder et al.*, 2022; *Zeng et al.*, 2022) that also observed no clear trend with plume age were examining Abs 365. We have added into the supporting information as Figure S4 the same series of plots as shown in Figure 10, but for the PILS water-soluble Abs 365 data from WE-CAN. A similar pattern as to what was observed at Abs 405 is seen for Abs 365. The text has also been updated to reflect this and Lines 497-500 now read as "As these three studies all examined Abs 365, the same series of plots shown in Figure 10 are repeated for PILS water-soluble Abs 365 and shown in Figure S4. A similar pattern was observed at both wavelengths for the WE-CAN data. This suggests our results were not wavelength specific and further corroborate the results observed during FIREX."

[Figure]

Figure S4. (a) PILS water-soluble Abs 365/WSOC, (b) PILS water-soluble Abs 365/$\Delta$CO, and (c) PILS water-soluble Abs 365/levoglucosan as a function of time since emission for all WE-CAN flights with the data segregated by flight.